# 3D Multi-Track and Multi-Layer Epitaxy Grain Growth Simulations of Selective Laser Melting

**DOI:** 10.3390/ma14237346

**Published:** 2021-11-30

**Authors:** Amir Reza Ansari Dezfoli, Yu-Lung Lo, M. Mohsin Raza

**Affiliations:** Department of Mechanical Engineering, National Cheng Kung University, Tainan 701401, Taiwan; ansari_amirreza@yahoo.com (A.R.A.D.); mmohsinraza80@gmail.com (M.M.R.)

**Keywords:** additive manufacturing, selective laser melting, epitaxy grain structure, cellular automaton, simulation and modelling

## Abstract

An integrated simulation framework consisting of the 3D finite element method and 3D cellular automaton method is presented for simulating the multi-track and multi-layer selective laser melting (SLM) process. The framework takes account of all the major multi-physics phenomena in the SLM process, including the initial grain structure, the growth kinetics, the laser scanning strategy, the laser–powder and laser–matter interactions, the melt flow, and the powder-to-liquid-to-solid transformations. The feasibility of the proposed framework is demonstrated by simulating the evolution of the epitaxy grain structure of Inconel 718 (IN718) during a 15-layer SLM process performed using a bi-directional 67° rotation scanning strategy and various SLM process parameters. The simulation results are found to be in good agreement with the experimental observations obtained in the present study and in the literature. In particular, a strong (001) texture is observed in the final component, which indicates that the grains with a preferred <001> orientation win the competitive epitaxy grain growth process. In addition, the size and shape of the IN718 grains are governed primarily by the cooling rate, where the cooling rate is determined in turn by the SLM parameters and the build height. Overall, the results show that the proposed framework provides an accurate approach for predicting the final microstructures of SLM components, and therefore, it can play an important role in optimizing the SLM processing parameters in such a way as to produce components with the desired mechanical properties.

## 1. Introduction

Selective laser melting (SLM) is a key technology for the fabrication of many components in the aerospace, automotive, and healthcare fields nowadays. Selective laser melting (SLM) provides customized designs, reduced preparation time, and the ability to create complicated shapes [1,2]. However, while SLM has the ability to produce both small- and large-scale components with extremely complex geometries, fabricating defect-free components remains a significant challenge [3,4,5]. Studies have shown that the majority of the defects found in parts, including a lack of fusion, porosity, and hot cracking, are related to the evolution of the microstructure during the build process [6,7]. Furthermore, the mechanical properties of even defect-free components are a function of their microstructural properties [8]. Rehman et al. developed a mechanism for detecting the flow behavior within the whole melt pool [9,10,11] that can predict melt flow inside the melting pool. However, developing a thorough understanding of the microstructural evolution of SLM components is essential in producing high-quality defect-free parts with the required mechanical properties and performance. The literature contains many experimental studies aimed at optimizing the SLM parameters in such a way as to improve the microstructure and mechanical properties of the built parts [12,13,14]. However, experimental approaches are time-consuming and expensive. Moreover, the SLM parameters exert both individual and interactive effects on the SLM outcome, and thus, experimental trial-and-error methods offer no guarantee of success in obtaining the optimal solution. As a result, the use of simulation modeling techniques to predict the microstructures of SLM components as a function of the processing conditions has attracted increasing interest in recent years [15,16].

Many studies apply the Phase Field (PF) and Cellular Automaton (CA) techniques to understand the effects of the microstructure grain evolution during the SLM process. In [17,18], the PF model has been discussed for different alloys used in SLM process, and the simulation results showed good agreements with experimental data. However, there are a few challenges that need to be addressed for the PF modeling as mentioned in [19], which include the lack of precursors, the absence of cluster–cluster interaction, diffusion controlled dynamics, large distance from metastable critical points, and the hard-sphere-like temperature scaling of the interfacial free energy.

The CA method, originally proposed by Gandin and Rappaz [20,21], takes into account all aspects of the solidification kinetics in SLM processing, including heterogeneous nucleation, the growth kinetics of the crystal structure, the preferred grain growth direction, and so on. Zinoviev et al. [15] used a two-dimensional (2D) CA model to evaluate the grain structure during the SLM processing. However, although the model successfully reproduced the columnar growth behavior of the grains during the build process, the final grain structure morphology was different from that observed experimentally due to the 2D nature of the simulation model. Lopez-Botello et al. [22] used an integrated 2D FE-CA model to investigate the grain structure evolution of alloys during laser melting. The simulation results provided useful insights into the effects of the powder–liquid–solid transformation process and multi-track and multi-layer interactions on the growth mechanics of the grain structure. However, as in [23], the final simulated grain structure was inconsistent with the experimental structure due to the use of a 2D model. Rai et al. [24] combined a 2D Lattice Boltzmann (LBM) thermal model with CA to simulate the grain structure formation during multi-layer powder bed fusion. The validity of the model was confirmed by comparing the simulation results for the grain structure of Inconel 718 (IN718) with the experimental findings for single track electron beam melting (EBM). In general, LBM is more accurate than FE. However, it has a high computational cost, and therefore, extending 2D LBM models to three-dimensional (3D) models in order to obtain more realistic simulation results is extremely challenging. Accordingly, Dezfoli et al. [25] used a 3D CA method instead to simulate the grain structure of Ti6Al4V after laser melting. The simulation results obtained for the solidification behavior of the molten alloy were shown to be in good agreement with the experimental observations. However, the model was limited only to single-track processing and was thus unable to accurately reproduce the evolution of the grain structure during realistic multi-track and multi-layer SLM processing. Koepf et al. [26] developed a 3D crystal growth model based on the CA method and an analytical thermal technique to simulate the grain structure of IN718 during laser powder bed fusion (L-PBF). It was shown that the simulation results for the grain structure were in good general agreement with the experimental grain structures of IN718 specimens fabricated by SLM. However, the model took no account of the effects of the melted and non-melted regions of the previous layer on the thermal history of the current layer. Consequently, there is a demand for an accurate simulation model that takes into account major physics and can simulate the grain structure with high accuracy.

Accordingly, the present study proposes an integrated framework based on a 3D FE-CA model to simulate the microstructural evolution of SLM components during a typical multi-track and multi-layer process. The model assigns different thermal properties to the bulk material and powder bed, respectively, and it treats the laser heat input as a 3D volumetric heat source that combines both laser-melt and laser-powder interactions. In addition, epitaxial grain growth, nucleation, and growth kinetics are considered in the 3D CA model. The feasibility of the proposed model is demonstrated by simulating the microstructural evolution of IN718 alloy during a 15-layer multi-track SLM process performed with various laser powers, scanning speeds, and hatch densities. The results confirm that the model accurately reproduces the final microstructure of SLM components following multi-track and multi-layer processing.

## 2. Model Description

Figure 1 presents a schematic illustration of the 3D computational domain considered in the present study, in which the specimen build direction is parallel to the Z-axis and the laser beam traverses in the XY plane and melts the powder, substrate, and previous layers, respectively. Since the main growth mode in the SLM processing of Ni super alloys such as IN718 is epitaxy growth [27,28,29], the simulations commenced by depositing an initial substrate consisting of 15 layers using a laser power, scanning speed, and hatch distance of 150 W, 800 mm/s, and 100 µm, respectively, and a bi-directional 67° rotation scanning strategy. Then, the last five layers of the built substrate (with a combined thickness of 200 µm) were used as the initial substrate for further SLM processing under various conditions.

As shown in the upper-schematic in Figure 1, the main SLM simulation process commenced by coating a powder layer on the substrate. The powder was treated as a continuous medium with properties determined from the powder size and powder pack density. Laser melting was applied to the powder layer using a bi-directional scanning strategy and the specified SLM processing conditions (i.e., the laser power, scanning speed, and hatch distance). The system was allowed to cool down until all the melt was solidified and a new powder layer was then deposited. The process was repeated in this way until a total of 15 layers had been deposited. For each layer, the simulations considered an epitaxy growth mode in which the partially re-melted grains in the previous layer (or the initial substrate in the case of the first layer) underwent a process of competitive epitaxy growth in accordance with their preferred growth directions. All of the layers in the built component remained computationally active until the end of the simulation process.

### 2.1. Nucleation and Growth Kinetics

The present simulations adopted the Gaussian model of continuous heterogenous nucleation presented by Thévoz et al. [30], in which the nucleation density is assumed to vary in accordance with the degree of undercooling, ∆*T*. In particular, the total nucleation density of the undercooled melt is given by [25,30]
(1)$$n\left(\Delta T\right)=\int_{0}^{\Delta T}\frac{n_{max}}{\sqrt{2\pi }}exp\left[-\frac{{\left(\Delta T-\Delta T_{max}^{0}\right)}^{2}}{2\Delta T_{\sigma }^{2}}\right]d\left(\Delta T\right)$$
where *n_max_* is the maximum density of nucleation; and $\Delta T_{max}^{0}$ and $\Delta T_{\sigma }$ are the mean and standard deviation of the undercooling degree, respectively. Based on this model, heterogenous nucleation was assumed to occur if the temperature of the melted cell fell in the range of ($T_{L}-\Delta T_{max}^{0}\pm \Delta T_{\sigma }$).

The undercooling temperature in Equation (1) is given by the summation of the thermodynamic and composition undercooling as
(2)$$\Delta T=\Delta T_{th}+\Delta T_{c}=T-T_{l}-m_{l}\left(C_{l}-C_{0}\right)$$
where *T_l_* is the equilibrium liquid temperature, *m_l_* is the liquidus slope, *C*_0_ is the initial equilibrium concentration, and *C_l_* is the solute concentration in the liquid phase. IN718 is considered here to consist only of Ni-19 at % Cr. In our model, the true multicomponent will not have too much of an impact on the results related to the growth kinetics of the CA model if two key elements for grain evolution in a specific alloy are chosen. For example, Ni and Cr as two major elements are chosen in IN718 alloy. Of course, for the sake of simplicity, a binary alloy in modeling is much easier than the true multicomponent IN 718 alloy.

In the present simulations, both epitaxy and bulk crystal growth were modeled using the Kurz–Giovanola–Trivedi (KGT) model [31]. Thus, the relationships between the degree of undercooling, ∆*T*, grain growth velocity, *v_g_*, and dendrite tip radius, *R_D_*, were given by
(3)$$\Delta T=-m_{l}C_{0}\left(1-\frac{1}{1-I_{v}\left(Pe\right)\left(1-k\right)}\right)$$
(4)$$R_{D}=\frac{2Pe\;D_{l}}{v_{g}}=2\pi \sqrt{\frac{\Gamma }{m_{l}G_{c}-G}}$$
where *k* is the Cr partitioning coefficient in IN718, *G* is the thermal gradient, *G_c_* is the solute gradient, *P_e_* is the Peclet number, and *I_v_* is the Ivantsov function, which is defined as [23,31]
(5)$$I_{v}\left(Pe\right)=\frac{C_{l}-C_{0}}{C_{l}\left(1-k\right)}=Pe\exp\left(Pe\right)\int_{Pe}^{\infty }\frac{e^{-\eta }}{\eta }d\eta .$$

By solving Equations (3)–(5), the growth velocity was obtained as [15,26]
(6)$$v_{g}=\frac{D_{l}}{5.51\pi^{2}{\left(-m_{l}\left(1-k\right)\right)}^{1.5}\Gamma }\left(\frac{{\left(T-T_{l}\right)}^{2.5}}{C_{0}^{1.5}}\right)$$
where Γ is the Gibbs–Thomson coefficient.

### 2.2. Heat Transfer

The SLM process uses a laser heat source to melt the powder bed at high temperature. Thus, having an accurate thermal model that takes proper account of the laser power, high-temperature phase change, and laser melt interactions, respectively, is essential. The thermal profile results taken from the heat transfer model were taken as one of the inputs to the CA model growth kinetics. It is noted that the heat transfer model is developed using COMSOL Multiphysics, and the CA code is written by authors. The governing equations for the SLM process include the energy equation, the Navier–Stokes equation, and the continuity equation, which are given respectively as [32,33]
(7)$$\frac{\partial \left(\rho h\right)}{\partial t}+\nabla \left(\left(\rho h\right)\vec{V}\right)=\nabla \left(k\nabla T\right)+{\dot{Q}}_{LP}+{\dot{Q}}_{LM}$$
(8)$$\rho \left[\frac{\partial \vec{V}}{\partial t}+\left(\nabla \vec{V}\right)\left(\vec{V}-{\vec{V}}_{ALE}\right)\right]=\nabla \left(-pI+\mu \left(\vec{\nabla }\vec{V}+{\left(\vec{\nabla }\vec{V}\right)}^{T}\right)\right)+S_{b}+S_{v}$$
(9)$$\frac{\partial \rho }{\partial t}+\nabla \;\left(\rho \vec{V}\right)=0$$
where *ρ* is the density, *T* is the temperature, $\vec{V}$ is the melt velocity vector, *μ* is the viscosity, *t* is time, *p* is pressure, and ${\dot{Q}}_{LP}$ and ${\dot{Q}}_{LM}$ are the volumetric laser heat sources applied to the powder bed and melt, respectively. In addition, *S_b_* is the buoyancy force, and it is given as [34,35,36]
(10)$$S_{b}=\rho \left(1-\beta \left(T-T_{m}\right)\right)\vec{g}$$
where *β* is the thermal expansion coefficient, *T_m_* is the melting temperature, and $\vec{g}$ is the gravity vector. The body force, *S_v_*, can be approximated using the Carman–Kozeny model as [34,35,36]
(11)$$S_{v}=-C\frac{{\left(1-f_{l}\right)}^{2}}{f_{l}^{3}+\varepsilon }\vec{V}$$
where *C* is a constant used to achieve high viscosity (i.e., zero flow) in the solid regions of the powder bed, and it is assigned a value of 10^6^ in the present study. In addition, *ε* is a small value used to prevent division by zero, and *f_l_* is the liquid fraction, which is defined as [35]
(12)$$f_{l}=\left\{\begin{array}{cc} 0 & T<T_{s}\\ \frac{T-T_{s}}{T_{l}-T_{s}} & T_{s}\leq T\leq T_{l}\\ 1 & T>T_{l} \end{array}\right\}$$
where *T_s_* and *T_l_* are the solidus and liquidus temperatures of IN718, respectively. From Equation (12), a value of *f_l_* = 0 represents the fully solid phase, while *f_l_* = 1 indicates a fully liquid phase.

As shown in Figure 2, the laser in the SLM process interacts with both the melt surface and the powder bed. In the present simulations, the laser–melt interaction was modeled in accordance with the following radial Gaussian distribution [37]:(13)$${\dot{Q}}_{LM}\left(r\right)=\frac{2P_{}}{\pi r_{0}^{2}}\left(1-R\right)\exp\left(-\frac{2r^{2}}{r_{0}^{2}}\right)$$
where *P* is the laser power, *R* is the reflection coefficient, *r*_0_ is the laser beam radius, and *r* is the distance of any point on the melt surface from the laser incident point. Meanwhile, the laser–powder bed interaction was modeled using the Beer–Lambert law, which takes into account the laser scattering effect between powder particles [38]. In particular, the laser heat source due to the laser–powder interaction is given by
(14)$${\dot{Q}}_{LP}\left(r\right)=\frac{2P}{\pi r_{0}^{*,2}}\left(1-R\right)\alpha \exp\left(-\frac{2r^{2}}{r_{0}^{*,2}}\right)\exp(-\int_{0}^{z}\alpha dl)$$
in which $r_{0}^{*}$ is the seat for the radius of effective laser interaction where, due to scattering effects between powder particles, $r_{0}^{*}$ > $r_{0}$, and *α* is the in-depth absorption coefficient.

The effects of latent heat during phase change were modeled as [35,39]
(15)$$c=\frac{\left(1-f_{l}\right)\rho_{s}c_{s}+f_{l}\rho_{l}c_{l}}{\rho }+\Delta H_{m}\frac{d\omega }{dT}$$
where ∆*H_m_* is the latent heat of fusion and *ω* is a smoothing function used to express the phase fraction during the phase change process, i.e., [38]
(16)$$\omega =\frac{f_{l}\rho_{l}-\left(1-f_{l}\right)\rho_{s}}{2\rho }.$$

The other thermal properties of the melt and bulk solid, i.e., the thermal conductivity, *k*, and density, *ρ*, were defined in accordance with the relative proportion of liquid and bulk solid phase, i.e., [39]
(17)$$k=\left(1-f_{l}\right)k_{s}+f_{l}k_{l}$$
(18)$$\rho =\left(1-f_{l}\right)\rho_{s}+f_{l}\rho_{l}$$
where *s* and *l* denote bulk solid and liquid properties, respectively. In practice, the powder bed in the SLM process is an equivalent medium composed of powder particles and gas-filled voids. Thus, the effective thermal conductivity of the powder layer can be described by the following Sih and Barlow model [40]:(19)$$k_{p}=k_{g}\left[\left(1-\sqrt{1-\varphi }\right)\left(1+\varphi \frac{k_{r}}{k_{g}}\right)+\sqrt{1-\varphi }\left(\frac{2}{1-\frac{k_{g}}{k_{r}}}\left(\frac{2}{1-\frac{k_{g}}{k_{r}}}ln\frac{k_{r}}{k_{g}}-1\right)\right)+\frac{k_{r}}{k_{g}}\right]$$
where *k_g_* is the gas thermal conductivity, *φ* is the porosity of the powder layer, and *k_r_* is the radiation thermal conductivity between the powder particles and is defined as [40]
(20)$$k_{r}=\frac{4}{3}\sigma T^{3}D_{p}$$
where *σ* is the Stefan–Boltzmann constant and *D_p_* is the average powder particle size. In the present simulations, the effects of heat radiation at the powder bed surface were modeled using the following effective emissivity [40]:(21)$$\varepsilon_{p}=\left(1-A_{H}\right)\varepsilon +A_{H}\varepsilon_{H}$$
where *A_H_* is the fraction of voids in the powder bed and *ε_H_* is the emissivity of the surface cavities, where these two terms are defined respectively as [40]
(22)$$A_{H}=\frac{0.908\varphi^{2}}{1.908\varphi^{2}-2\varphi +1}$$
(23)$$\varepsilon_{H}=\frac{\varepsilon \left[2+3.082{\left(\frac{1-\varphi }{\varphi }\right)}^{2}\right]}{\varepsilon \left[1+3.082{\left(\frac{1-\varphi }{\varphi }\right)}^{2}\right]+1}.$$

All of the other physical properties of the powder bed (denoted generically as ∅*_p_*) were expressed simply as a function of the corresponding bulk IN718 properties, ∅, and porosity of the powder layer, *φ*, i.e., [41]
(24)$$\emptyset_{p}=\left(1-\varphi \right)\emptyset .$$

Radiation and convection heat transfer boundary conditions were imposed on the top surface of the melt pool and powder bed. In addition, Marangoni and surface tension boundary conditions were also imposed on the melt surface, i.e.,
(25)$$\sigma =-\gamma n\left(\nabla .n\right)+\frac{\partial \gamma }{\partial T}\left(\nabla T-\left(\nabla T.n\right)n\right)$$
where *n* is the unit vector normal to the melt surface and *γ* is the surface tension coefficient.

### 2.3. Coupling of CA and Heat Transfer Model

The computational domain of the CA model was partitioned into cubic cells, where each cell was assigned three variables, namely (1) the temperature, (2) the state, and (3) the crystallographic orientation. The temperature was calculated from the heat transfer model and interpolated to each of the CA cells in accordance with the specified time. The state of each cell was set as either powder (−1), solid (1), or liquid (0). In the initialization stage, all of the cells in the initial substrate region of the domain were assigned a state of 1, while all the cells in the powder layer were assigned a state of −1. During the subsequent simulation process, the states of the solid and powder layer cells were changed to 0 (i.e., the liquid state) if the temperature exceeded the liquidus temperature. As shown in Figure 3, based on previous experimental observations [40], the lattice structure of IN718 was assumed to be face-centered cubic (fcc) and to maintain an octahedral grain morphology during growth. During cooling, each solid cell lies at the solid–liquid interface when its temperature passes through the liquidus temperature and starts to grow with a preferential direction determined by its crystallographic orientation (as represented in 3D space by three Euler angles). During the growth process, the growth of each solid cell, i, at the solid–liquid interface was described by [28]
(26)$$l_{i}^{j}\left(t_{c}\right)=W_{i}^{j}\int_{0}^{t_{c}}v_{g}dt$$
where *v_g_* is the growth speed obtained from the KGT model in Equation (6), and $W_{i}^{j}$ is the orientation weight coefficient, which is related to the angle between the preferential growth angle of cell *i* and the vector leading from cell *i* to cell *j* (referred to as vector $L_{i}^{j}$) [42]. The orientation weight coefficient, $W_{i}^{j}$, is formally defined as [43]
(27)$$W_{i}^{j}=Max\left[\left|X_{w}\right|,\left|Y_{w}\right|,\left|Z_{w}\right|\right]$$
where *X_w_*, *Y_w_*, and *Z_w_* are calculated as [40]
(28)$$\left[\begin{array}{c} X_{w}\\ Y_{w}\\ Z_{w} \end{array}\right]=\left[\begin{array}{ccc} x_{p} & x_{q} & x_{r}\\ y_{p} & y_{q} & y_{r}\\ z_{p} & z_{q} & z_{r} \end{array}\right].\left[\begin{array}{c} \begin{array}{c} p_{i}^{j}\\ q_{i}^{j} \end{array}\\ r_{i}^{j} \end{array}\right]$$
in which (*x_p_*, *x_q_*, *x_r_*), (*y_p_*, *y_q_*, *y_r_*), and (*z_p_*, *z_q_*, *z_r_*) are the direction cosines of the [100], [010], and [001] dendrite arms with respect to the x-, y-, and z-axes of the coordinate system, respectively. In addition, ($p_{i}^{j}$, $q_{i}^{j}$, $r_{i}^{j}$) are the direction cosines of vector $L_{i}^{j}$ relative to the x-, y-, and z-axes, respectively [25]. Finally, the liquid cell *j* is considered to be captured by the solid cell *i* if
(29)$$\frac{l_{i}^{j}\left(t_{c}\right)}{L}=1$$
where *L* is the distance between cell *i* and cell *j*. In practice, *L* = *l_cell_* if *j* is one of the six nearest liquid neighbors of cell *i*, *L* = $\sqrt{2}$*l_cell_* if *j* is one of the twelve second-nearest neighbors, and *L* = $\sqrt{3}$*l_cell_* if *j* is one of the eight third-nearest neighbors. After capturing cell *j* by growing cell *i*, the state of cell *j* is changed to the state of cell *i* and assigned the same crystallographic orientation.

### 2.4. Simulation Parameters

As shown in Figure 1, the heat transfer model was implemented in a rectangular domain with a length and width of 2000 µm and a height that varied in accordance with the number of deposited layers. For all of the simulation cases considered in the present study, the powder layer thickness and powder layer porosity were set as 40 µm and 40%, respectively, while the initial substrate thickness was set as 200 µm. In addition, the lower surface of the substrate was maintained at a constant temperature of 293.5 K throughout the entire SLM process, and the 500 µm × 500 µm center region of the heat transfer model was assigned to the CA model with a uniform cell size of 1 µm × 1 µm × 1 µm. The initial temperature of each new powder layer was assumed to be 293.5 K, and heat loss radiation and convection boundary conditions were applied on all the system surfaces. In addition, the time step, *τ*, was selected adaptively in such a way that the maximum growth rate in each time step was maintained at ¼ of *l_cell_*, i.e.,
(30)$$\tau =\frac{l_{cell}}{4v_{g,max}}.$$

The time step for the FE heat transfer model was kept below that of the CA time step using the following stability criterion [23]:(31)$$\tau =\frac{\rho c_{p}h^{2}}{6\lambda }$$
where *h* is the mesh length. Table 1 lists all of the other thermal and microstructural parameters used in the simulations.

## 3. Results and Discussion

### 3.1. Thermal Model Validation

The basic validity of the proposed heat transfer model was evaluated by comparing the simulation results for the melt pool size of IN718 in single-track SLM processing with the experimental results for various values of the heat input intensity (defined as the ratio of the laser power (*P*) to the scanning speed (*v*) [46]) and a constant powder layer thickness of 40 µm. Figure 4 compares the two sets of results for the melt pool width and melt pool depth for heat input intensities in the range of 22 to 38 J/m. Note that for the experimental samples, the single-track specimens were ground, mechanically polished, and etched, and then, they were observed using optical microscopy (OM) to obtain the melt pool dimensions. Both the simulation results and the experimental results show that the melt pool size increases with an increasing heat input, i.e., an increasing laser power or a decreasing scanning speed. Moreover, the simulation results deviate from the experimental results by no more than 12%. Hence, the basic validity of the heat transfer model is confirmed. Moreover, it is is noted that in our model and experimental validation, only the conduction mode of the melt pool is taken into the account, and the keyhole and balling effects due to the convection mode will be considered in the future.

### 3.2. Microstructural Evolution and Validation

The validity of the coupled FE–CA model was investigated by comparing the simulation results for the grain structure evolution during multi-track and multi-layer SLM processing with the experimental observations. A 15-layer SLM process was simulated using a laser power of 150 W, a scanning speed of 800 mm/s, a hatch distance of 100 µm, and a bi-directional scanning strategy with a 67° rotation angle between layers. For comparison purposes, snapshots of the simulated grain structure were obtained after each of the final eight layers in the build process. Then, experimental trials were performed under the same SLM conditions using a commercial SLM machine (EOS M290, EOS GmbH, Krailling, Germany) with an argon-filled chamber for an oxygen concentration of less than 2000 ppm to prevent oxidation. Following the printing process, the built part (a simple rectangular component) was sectioned using an EDM wire cutting machine, polished by hand, and then further polished on an ion polishing machine to remove any last remaining scratches prior to electron backscattered diffraction (EBSD) (ZEISS Supra 55, ZEISS, Oberkochen, Germany) analyses of the grain size, grain shape, and texture.

Figure 5a–h show the simulation results obtained for the 3D grain structures in the final eight layers of the built component. Figure 5i shows the EBSD results for the final component. Finally, Figure 5j,k show the pole figures of the final simulated grain structure, where the different colors represent different orientation angles between the build direction (z) and the crystal [001] axis.

Note that in the simulation snapshots presented in Figure 5a–h, the melt and powder regions are deliberately not distinguished in order to improve the visibility of the grain structure. However, the boundaries of the individual layers are shown in yellow dashed lines for visualization purposes. The simulation and experimental results in Figure 5 both show the presence of a columnar grain growth mode. In particular, an elongated grain morphology is observed in a direction approximately parallel to the z-axis in the XZ and XY planes, while a uniform 2D equiaxed structure is seen in the XY plane. The two sets of results are not only in good qualitative agreement with one another in all three planes but are also consistent with those reported in the literature, which proved that the columnar form of the grain structure forms after the SLM process [48,49,50]. Thus, the validity of the FE–CA model is confirmed. In this current study, only the qualitative analysis has been made for the multi-layer and multi-track epitaxial grain growth during the SLM process, and the quantitative study will be carried out in the future.

The formation of a columnar grain shape in a direction perpendicular to that of laser movement is to be expected since grain growth follows the temperature gradient [40]. During SLM, most of the heat is extracted from the work piece via heat conduction from the lower region of the melt pool toward the substrate. Consequently, a high thermal gradient is induced in the build direction, and thus, columnar grain growth occurs. The results presented in Figure 5 imply that the heat extraction from the lower region of the melt pool provides sufficient kinetic force for the partially melted grains in the previous layer (or initial substrate) to grow into melt while keeping their original orientation. Thus, the length of the columnar grains exceeds the powder layer thickness, as shown in Figure 5a–g.

The presence of so many partially re-melted grains in the built part leads to a competitive grain epitaxial growth process, in which the grains with a crystallographic orientation parallel to the thermal gradient direction grow faster and win the competition. The effects of this competitive grain growth mechanism can be observed in the pole figures shown in Figure 5j,k, wherein a clear crystallographic texture is discerned with a strong (001) orientation in the final microstructure.

Further insights into the grain structure were obtained by calculating the grain size based on the number of CA cells belonging to each grain, i.e.,
(32)$$V_{eq,G}=\left(number\;of\;CA\;cells\right)\times dl_{CA}^{3}$$
(33)$$d_{G}=2\sqrt[{3}]{\frac{3V_{eq,G}}{4\pi }}.$$

Having determined the size of each individual grain, the average grain size of the sample was evaluated as follows:(34)$$d_{G,ave}=\frac{\sum_{i=1}^{N_{G}}d_{G}}{N_{G}}$$
where *N_G_* is the total number of grains taken into account in the calculation process. Figure 6 presents the statistical grain size distribution obtained from the simulation results given in Figure 5. As shown, the average grain size is equal to 18.8 µm, while the majority (≈76%) of the grains have a size of less than 20 µm.

To gain more insights into the competitive grain growth process, two grains, one with a small (5°) preferred orientation angle relative to the Z-axis (grain A) and one with a large (43°) preferred orientation angle relative to the Z-axis (grain B) were selected, and their final shapes after the SLM process were examined, as shown in Figure 7. It is seen that both grains experience epitaxy growth from a root in the substrate while grain B has a wider root than grain A. Moreover, due to its larger disorientation angle, grain B tends to grow in the lateral direction rather than the build direction. Consequently, the grain growth direction is poorly aligned with the thermal gradient, and the grain loses the competitive grain growth process. By contrast, grain A grows more rapidly in the build direction (i.e., the thermal gradient direction). Thus, as shown in Figure 7, the growth of grain B is truncated, whereas that of grain A continues right up to the surface of the built part. The fact that grain B grows only to the 7th layer before truncation implies that grain selection during SLM processing with a continuous bi-directional rotating strategy takes only a very short time. In other words, the truncation of unfavorable grains occurs within a distance of just several hundred microns. Interestingly, the results presented in Figure 7a show that the lateral size of grain A remains approximately constant after seven layers, which suggests that due to the presence of only preferential grains after the first few hundred microns of SLM processing, the competitive grain process is replaced by a simple equal-competition grain growth mechanism.

### 3.3. Microstructural Analysis Results for Different SLM Process Parameters

To investigate the effects of the SLM processing conditions on the grain structure of the IN718 components, further simulations were performed using 12 different combinations of the laser power (150 W and 200 W), laser scanning speed (800 and 900 mm/s), and hatch distance (80, 90, and 100 µm), resulting in energy densities ranging from 49.4 to 78.1 J/mm^3^. Note that the energy density was computed as
(35)$$E=\frac{P}{vHL_{t}}$$
where *v* is the laser scanning speed, *H* is the hatch distance, and *L_t_* is the powder layer thickness with a constant value of 40 µm. For each considered set of simulation conditions, 15 layers were deposited on the initial substrate, and the calculation results were limited to the last 10 layers.

Table 2 shows the calculation results obtained for the average grain size under each of the considered energy densities. As expected, the average grain size increases with an increasing energy density. In other words, the grain size increases with a reducing hatch distance and laser scanning speed and an increasing laser power. The grain coarsening effect with an increasing energy density is caused by a local altering of the cooling rate. In particular, as the laser energy density increases, the temperature of the powder bed around the melt pool also increases. This decreases the local cooling rate and prompts more intense competitive epitaxy grain growth. Consequently, strong epitaxy grain growth along the preferred grain orientation (i.e., the build axis) occurs, and therefore, the grains capture more melt and become larger as a result.

### 3.4. Effects of Cooling Rate on Grain Size

Figure 8 shows the simulation results obtained for the variation of the average grain size with the cooling rate during SLM single-track processing with a constant laser power of 200 W and various values of the hatch distance (80, 90, and 100 µm) and laser scanning speed (800 and 900 mm/s). It is seen that the cooling rate decreases with a reducing hatch distance and decreasing scanning speed. In other words, the cooling rate decreases with an increasing energy density. Additionally, it is seen in Figure 8 that the largest grain size (19.5 µm) is associated with the lowest cooling rate 2.3 × 10^5^ (K/s). Hence, it can be inferred that the grains undergo a coarsening effect with an increasing laser energy density due to the lower cooling rate.

To gain further insights into the change of the cooling rate during the layer-by-layer printing process, the cooling rate was monitored at the center point of the computational domain in layers 1, 3, and 5, respectively, as shown in Figure 9. The results show that the cooling rate is dependent not only on the SLM parameters (i.e., the hatch distance, laser scanning speed and laser power) but also on the distance of the layer from the substrate (build height). For example, a detailed inspection of Figure 9 shows that the maximum cooling rate decreases from ≈12 × 10^5^ K/s to 6.1 × 10^5^ and 4.2 × 10^5^ K/s as the build height increases from one to three to five layers, respectively. In other words, the heat loss reduces gradually as the build process proceeds. This finding is intuitive, since in the initial stage of the SLM process, the cooling rate is enhanced as a result of the rapid conductive heat transfer from the melt pool to the substrate. However, the cooling rate decreases as the distance from the melt pool to the substrate (i.e., the heat sink) increases. Under a lower cooling rate, the intensity of the competitive grain growth process is reduced. Thus, the grains tend to reach a constant size at a greater build height, as shown earlier in Figure 7 for grain A.

### 3.5. Predication of Pores

The presence of various pores weakens the bonding strength between adjacent layers and therefore may lead to mechanical failure [13,49]. Specially, the lack of fusion in pores has an irregular shape that is not circular, and this results in stress concentration, while the components go under loading [46]. In the present study, the porosity of the SLM components built under different processing conditions was evaluated by determining the ratio of the CA cells, which underwent no phase change during the build process to the total number of CA cells in the computational domain. The corresponding results are presented in Table 3. The results show that the porosity reduces under a higher energy density and a lower hatch distance. For example, given an energy density of 55.5 J/mm^3^, the porosity reduces from 2.9 to 1.3% as the hatch distance decreases from 100 to 80 µm. In other words, to produce components with low porosity, increasing the energy density alone is insufficient. In general, reducing the energy density results in a smaller melt pool size given a larger hatch distance and hence increases the likelihood of the formation of pores. Overall, the results suggest that by using a 67° rotation angle between layers, dense components can be obtained using a smaller hatch distance. It is noted that the present results are consistent with those of previous studies on the dependency of the formation of pores on the laser energy density, which showed that both the sample density and the mechanical properties are functions of the laser scanning speed and hatch distance [51,52].

Figure 10 shows the image taken from a 3D cube after polishing for checking the porosity at an energy density of 55.5 J/mm^3^ and hatch distance of 80 µm. As a result, the porosity density of 0.011% is calculated, and its corresponding predicted porosity density in Table 3 is 1.30%. In the predication of porosity density illustrated in Table 3, it is noted that the global trend of predication in porosity density using the CA model is quite reasonable qualitatively. However, the quantitative predication in porosity density is quite difficult because many factors in the experimental design will significantly affect the porosity density. In addition, those uncertain factors are strongly dependent upon the SLM machine itself.

## 4. Conclusions

This study has presented a combined 3D finite element-cellular automaton framework for simulating multi-track and multi-layer SLM processes. It has been shown that the simulation results obtained for the grain structure and final texture of the IN718 component are in good agreements with the experimental observations. The simulation and experimental results support the following major conclusions:An equiaxed grain structure is formed in the XY plane of the printed part in a direction perpendicular to the build direction. In addition, elongated grains with a length extending through several layers are formed in the vertical XZ and YZ planes.The final SLM component has a strong (001) texture, which indicates that most of the grains that win the epitaxy competitive grain growth process undergo preferred orientation in the <001> direction relative to the build direction (i.e., the z-axis).The grain size in the built part is directly related to the cooling rate. In particular, a lower cooling rate gives rise to a more uniform grain structure. The cooling rate can be actively controlled by adjusting the SLM process parameters such as the laser power, scanning speed, and hatch distance, and it also decreases naturally as a function of the build height.The density of the lack of fusion pores formed in the built component reduces with an increasing energy density and a reducing hatch distance.

Overall, the model presented in this study provides useful insights into the effects of the SLM process parameters on the final microstructure of the printed part and therefore serves as a useful tool in tuning the SLM parameters in such a way as to produce SLM parts with the required microstructural properties and mechanical behavior. In the future, researchers can use a similar approach as introduced in [53,54] for an in situ observation through Machine Learning for microstructure modeling, which can help to control the final properties for SLM manufactured parts.

## Figures and Tables

**Figure 1 materials-14-07346-f001:**
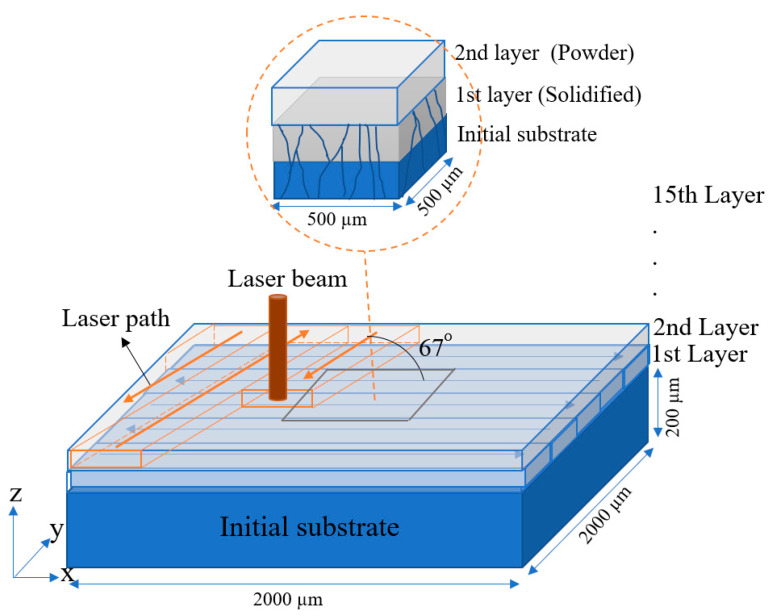
Schematic illustration of computational domain for SLM simulations.

**Figure 2 materials-14-07346-f002:**
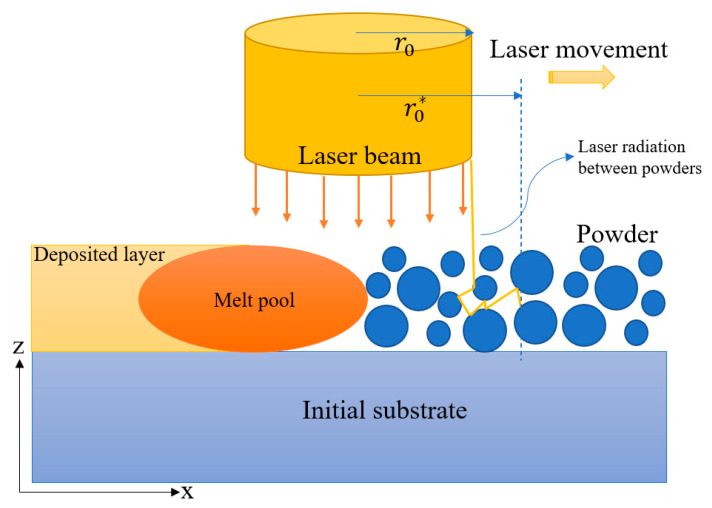
Schematic illustration of laser interaction with melt pool and powder particles.

**Figure 3 materials-14-07346-f003:**
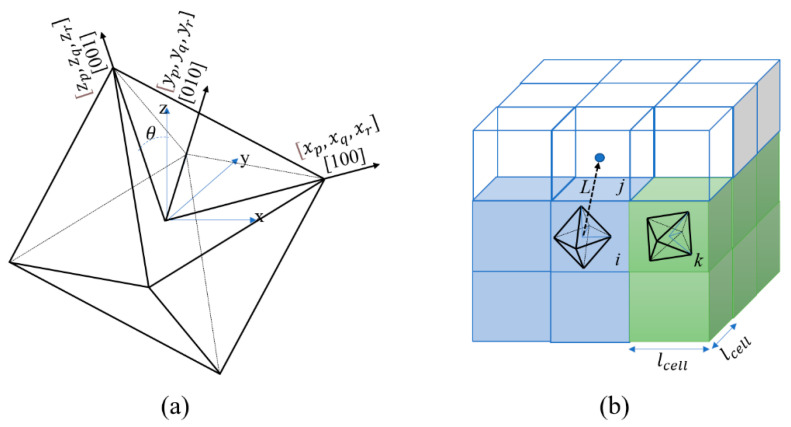
(**a**) Schematic illustration of 3D octahedron grain model in the CA scheme. (**b**) Three-dimensional (3D) view of two solid cells *i* and *k* and their crystal orientations relative to the computational domain during the capturing of upstream liquid cells.

**Figure 4 materials-14-07346-f004:**
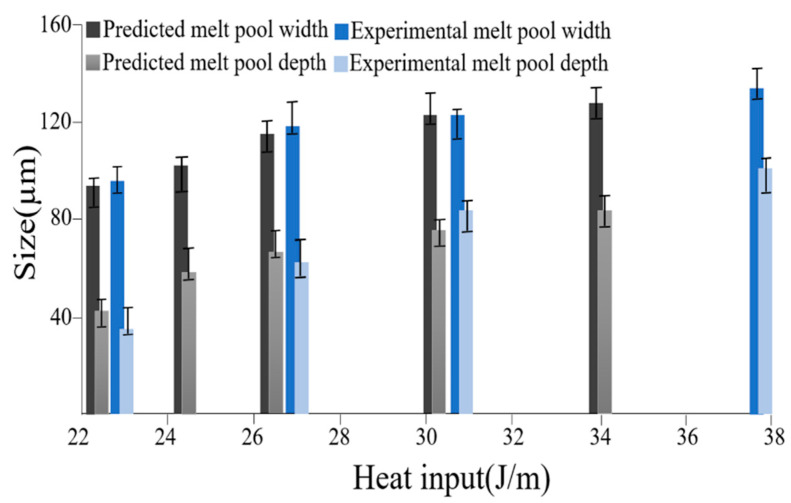
Simulation and experimental results for melt pool size given various values of the heat input intensity.

**Figure 5 materials-14-07346-f005:**
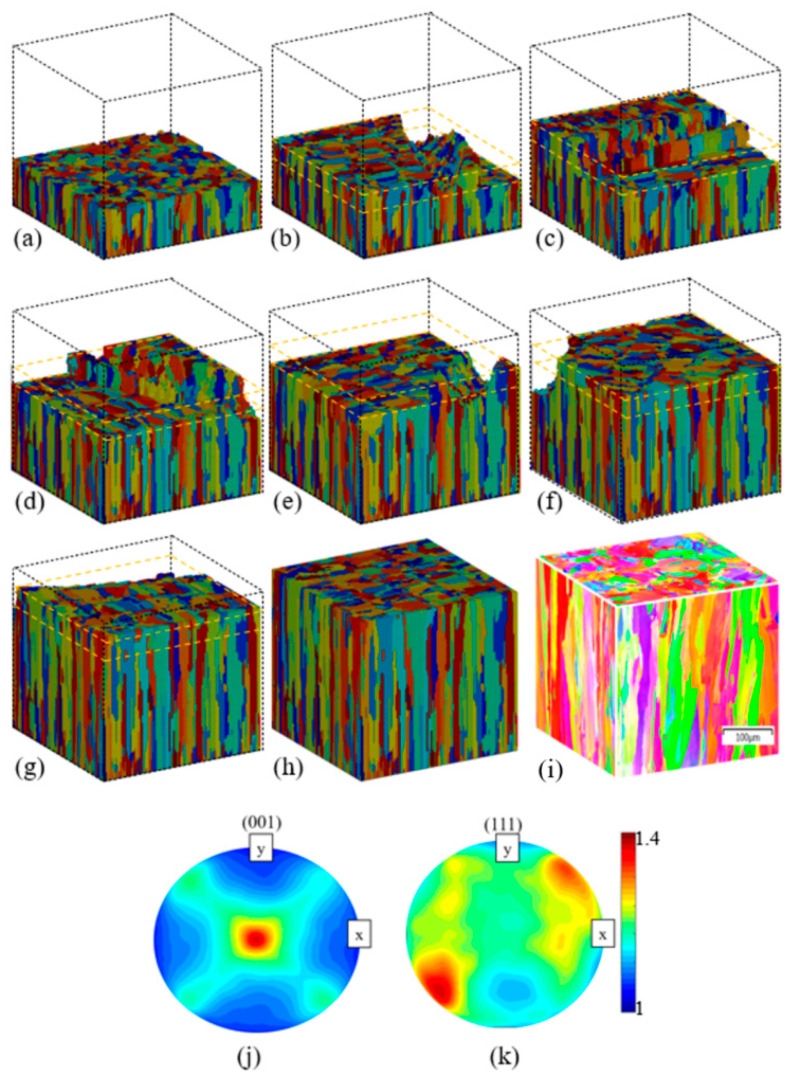
Grain structure evaluation of last eight layers (8th to 15th) in the SLM built part (**a**) 8th layer, (**b**) 9th layer, (**c**) 10th layer, (**d**) 11th layer, (**e**) 12th layer, (**f**) 13th layer, (**g**) 14th layer, and (**h**) 15th layer. (**i**) EBSD analysis results for built SLM component. (**j**,**k**) Pole figures of simulated grain structure.

**Figure 6 materials-14-07346-f006:**
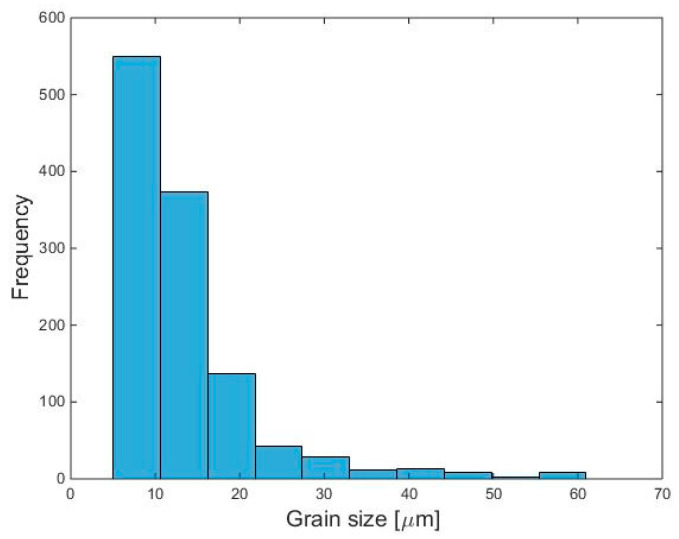
Statistical results for simulated grain size distribution in IN718 built component.

**Figure 7 materials-14-07346-f007:**
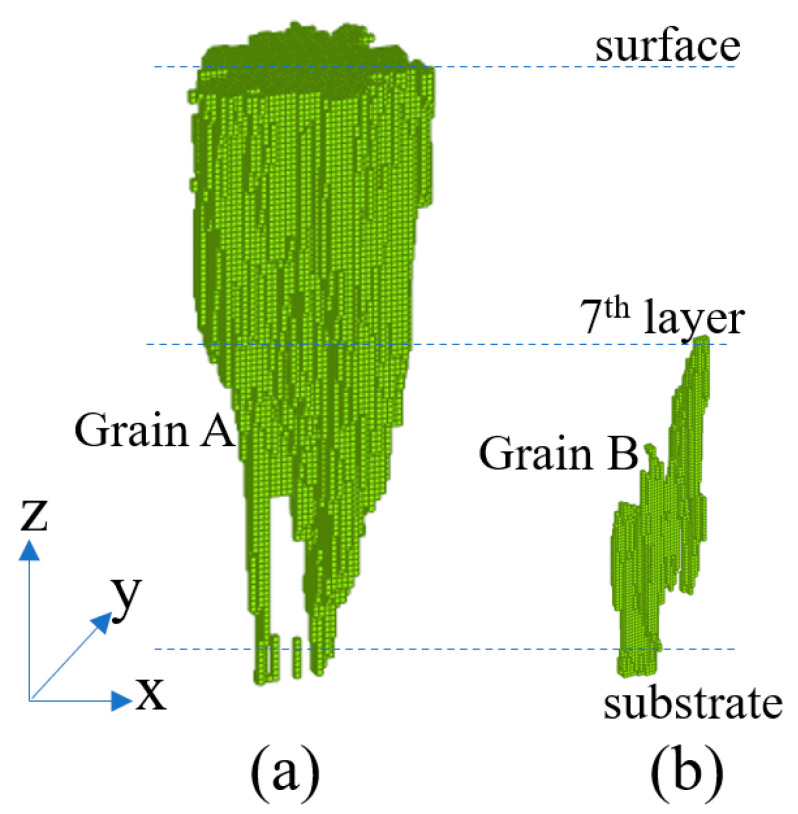
Shape comparison of grains with different disorientation angles relative to the build direction: (**a**) grain A, 5° disorientation, and (**b**) grain B, 43° disorientation.

**Figure 8 materials-14-07346-f008:**
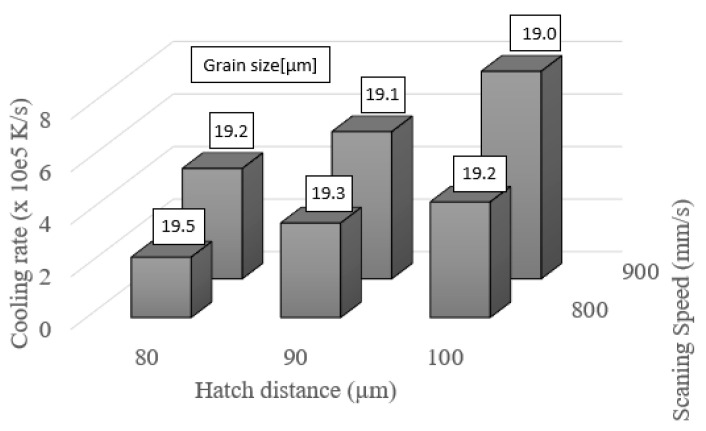
Relation between SLM process parameters and cooling rate. (Note that the average grain size is shown in the boxes above the bars.).

**Figure 9 materials-14-07346-f009:**
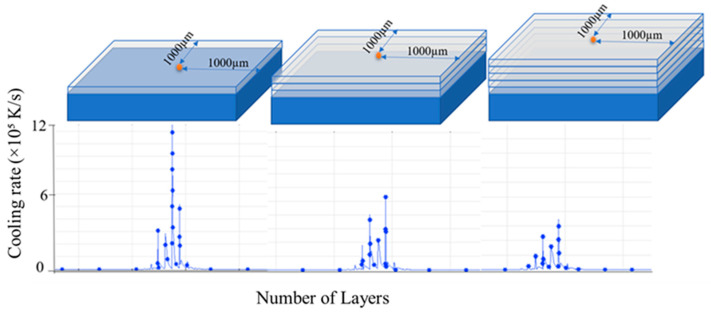
Cooling rates at center points of layers located at different build heights.

**Figure 10 materials-14-07346-f010:**
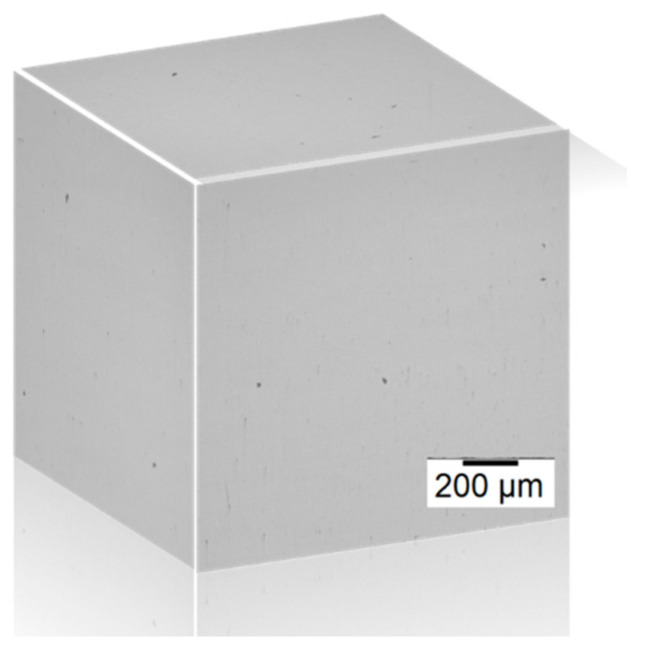
Three-dimensional (3D) OM image of cubic sample after polishing for porosity analysis.

**Table 1 materials-14-07346-t001:** Parameters used in FE–CA simulations.

Property (Unit)	Nomenclature	Value	Reference
Liquidus temperature [°C]	*T_l_*	1336	[44]
Solidus temperature [°C]	*T_s_*	1260	[44]
Evaporation temperature [°C]	*T_v_*	2911	[45]
Density of liquid [kg.m^3^]	*ρ* _l_	7300	[46]
Density of solid [kg.m^3^]	*ρ_s_*	8190	[46]
Conductivity of liquid [J/m.s.K]	*k_l_*	29.3	[44]
Conductivity of solid [J/m.s.K]	*k_s_*	−7.0 × 10^−6^ T^2^ + 0.0294 T + 0.5603	[44]
Specific heat of liquid [J/kg.K]	*c_l_*	720	[46]
Specific heat of solid [J/kg.K]	*c_s_*	512	[46]
Latent heat of fusion [kJ/kg]	Δ*H_m_*	270	[44]
Latent heat of vaporization [kJ/kg]	Δ*H_v_*	6690	[44]
Viscosity [kg/ms]	µ	7.8 × 10^−3^	[45]
Surface tension [N/m]	*γ*	1.89	[45]
Marangoni coefficient [N/m.K]	∂γ/∂T	−1.1 × 10^−4^	[45]
Absorption (liquid) [mm^−1^]	*A*	25	[45,46]
Reflection coefficient	*R*	0.7	[46]
Liquidus slope [K/wt %]	*m_l_*	−10.5	[47]
Initial equilibrium concentration [wt %]	*C* _0_	0.5	[47]
Gibbs-Thomson coefficient [K.m]	Γ	3.65 × 10^−7^	[48]
Solute diffusion coefficient [m^2^/s]	*D_l_*	3 × 10^−9^	[48]

**Table 2 materials-14-07346-t002:** Average grain size in SLM parts printed with different process parameters and bi-directional scanning strategy with 67° rotation between layers.

Energy Density (J/mm^3^)	Laser Power (W)	Scanning Speed (mm/s)	Hatch Distance (µm)	Average Grain Size (µm)
55.5	150	900	80	18.7
49.4	150	900	90	18.3
62.5	150	800	80	19.0
55.5	150	800	90	18.9
50	150	800	100	18.8
69.4	200	900	80	19.2
61.7	200	900	90	19.1
55.5	200	900	100	19.0
78.1	200	800	80	19.5
69.4	200	800	90	19.3
62.5	200	800	100	19.2

**Table 3 materials-14-07346-t003:** Average size of lack of fusion pores for different SLM process parameters and bi-directional scanning strategy with 67° rotation angle between layers.

Energy Density (J/mm^3^)	Laser Power (W)	Scanning Speed (mm/s)	Hatch Distance (µm)	Porosity %
55.5	150	900	80	1.30
49.4	150	900	90	7.30
62.5	150	800	80	1.10
55.5	150	800	90	2.90
50	150	800	100	3.60
69.4	200	900	80	0.09
61.7	200	900	90	0.77
55.5	200	900	100	1.80
78.1	200	800	80	0.03
69.4	200	800	90	0.19
62.5	200	800	100	0.24

## Data Availability

The data presented in this study are available on request from the corresponding author.

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
