# Peer review of "3D Multi-Track and Multi-Layer Epitaxy Grain Growth Simulations of Selective Laser Melting"

_materials, 2021, doi:10.3390/ma14237346_

Round 1

Reviewer 1 Report

-Introduction should (briefly) discuss other microstructural solidification modeling techniques as well, such as phase field models, and the trade-offs between these techniques.

-How would the results change if in the CA growth kinetics you wouldn't assume a binary alloy, but the true multicomponent IN718 alloy? 

-How does this CA model deal with spurious anisotropy in the texture (see e.g. https://doi.org/10.1016/j.actamat.2021.116930 )? How is the growth anisotropy magnitude in Equation 26 and 27 quantified? Can the growth anisotropy strength be altered? How does this relate to capillary anisotropy (typically having values 0.01-0.05 for FCC metals) or kinetic anisotropy? The growth anisotropy is expected to strongly affect the texture, and its effect should be discussed (including the assumptions behind the used growth anisotropy).

-Table 1: CA and heat transfer model parameters should be separated in the table for clarity.

-Figure 9 should be made easier to read: the relation between the cooling rate graph X-axis and the arrows is somewhat hard to read.

-Equation 7-9 should have references, Eq. 7 left-hand-side should have divergence operator, not gradient.

-The CFD model as such does not take into account how evaporation + recoil pressure affect the melt flow, so keyholing cannot be properly predicted. This should be mentioned in the manuscript. For the process parameters considered, is the melt pool expected to operate in conduction mode, meaning without keyholing (due to evaporation + recoil pressure effects)? 

-It seems that the time step size is not limited by fluid flow related effects, such as CFL condition, but only the linear stability criterion for heat conduction. Is e.g. the CFL condition not violated in the simulations?

-What is the mesh length for the heat transfer modeling? Comment on the heat transfer model convergence with respewct to mesh length.

-Figure 4: error bars should be given. Are the melt pools in conduction mode in the experiments?

-Figure 5 only qualitatively compares the experiments to the simulations. Why is the grain sizes not estimated from experiments, or their pole figures not shown? The comparison is shown only to be qualitatively correct currently (lines 291-301), and this should be indicated in the abstract and conclusions (or alternatively a quantitative comparison should be performed).

-Figure 6: there should be another metric as well to describe the grain size distribution, which looks power-law-like. How do the grain size distributions compare to literature (simulations or experiments)?

-Was the pore size estimated? Were they individual cells/voxels or larger domains?

Author Response

Authors gratefully acknowledge with thanks of receiving the chance of manuscript content revision with useful comments from the reviewers. We have carefully made the revision of this manuscript in the following attachment. Our answers to each comment in the revised manuscript are indicated in Palatino Linotype marked yellow highlight tool.

Reviewer 2 Report

The present study utilized three-dimensional finite element modeling and cellular automaton to explore the grain growth behavior in the SLM of Inconel 718. The effects of multi-layers and processing conditions on grain growth behavior were explored.

Although the manuscript was well-written and the purpose of the manuscript was clear, there were points that would make the study more comprehensive if authors could include/discuss.

  1. Did authors develop their own FEM/CA code or utilize the commercial packages? It would be nice if this point can get clarified.
  2. A thermal simulation was indirectly validated by comparing with experimental melt pool dimensions. It was not clear to me if authors have conducted their own test or referred to results of existing literature. If it is former, please kindly include the images from the experiments.
  3. It was not clear to me how the grain size prediction from the CA model was affected by the presence of lack of fusion pores, especially at the low energy density. Could authors elaborate on this point?
  4. There is a typo at line 379. I believe it’s meant to be “2.3 x 105 (K/s)”
  5. The difference in max and min grain size for all cases as seen in Table 2 was quite small (~ 5%). Did that imply the process conditions only had a minor effect on microstructures?
  6. Followingly, I wish authors could elaborate the implication of process-dependent grain sizes on mechanical behaviors, and if the in-situ mechanical property control is possible in the SLM of Inconel 718 based on what authors see from the simulation results.
  7. In section 3.5, please provide more details on the lack of fusion comparison. There were no images from both experiments and simulation in this section. In my opinion, including these figures can covey more clearly the capability of the numerical model developed by the authors.
  8. Also, please provide more details on which measurement techniques were used to assess the porosity
  9. The conclusion was quite long, and many well-known facts were stated. Please focus on the novel finding from the study for better clarity and concision

Author Response

(The authors gave the same response as above.)

Reviewer 3 Report

The authors present an integrated simulation framework for simulating the multi-track and multi-layer selective laser melting process. This work lies within an imporatnt area of R&D at the moment and sheds light into microstructure development in AM - which is hard to observe. It may be considered for publication, if the authors address the following:

SPECIFIC COMMENTS

  1. The authors must clearly state that the FE model is used to obtain the temperature histories which are passed on to the CA model which the computes the microstructures.
  2. The authors must discuss in some details the limitations of their approach. For instance, the FE model cannot simulate fluid flow (and hence the Marangoni effects) that influence phenomena that happen at the microstructure level which are relevant to microstructure formation. Thus, FE modelling is considered somewhat inferior to Computational Fluid Dynamics (CFD) models for this purpose. Also, CA models ignore some microscopic phenomena to provide a speedier solution. These must be discussed at least briefly. 
  3. The authors must take care to use correct language. For instance, in the sentence located at P1 L33-35 is NOT accurate in that the lack of fusion defect is not necessarily determined by microstructure formation (but by heat transfer related phenomena). Rather, the microstructure can be influenced by a lack of fusion (through temperature effects). A stronger effort must be be made for delivering the correct message to the reader. It is worth checking the manuscript carefully for other such instances.
  4.  The authors have done a satisfactory literature review. If they wish to update their review with more recent work, they can look up: https://www.mdpi.com/journal/metals/special_issues/simulation_evolution 

Author Response

(The authors gave the same response as above.)

Round 2

Reviewer 2 Report

I appreciate author's effort to address my comments as suggested. 

Author Response

The authors would like to thank the reviewer for the insightful and constructive comments.